# Microstructure and Mechanical Properties of Laser Direct Energy Deposited Martensitic Stainless Steel 410

**DOI:** 10.3390/mi15070837

**Published:** 2024-06-28

**Authors:** Hyun-Ki Kang, Hyungsoo Lee, Chang-Seok Oh, Jongcheon Yoon

**Affiliations:** 1R&D Center, Turbo Power Tech, #107 Dasan-ro, Saha-gu, Busan 49488, Republic of Korea; 2Korea Institute of Materials Science, 66 Sangnam-dong, Changwon 51508, Republic of Korea; 3Customized Manufacturing R&D Department, KITECH, 113-58 Seohaean-ro, Siheung 15014, Republic of Korea

**Keywords:** additive manufacturing, directed energy deposition, martensitic stainless steel 410, chrome carbide, tempering

## Abstract

The aim of this work is to study the phase transformations, microstructures, and mechanical properties of martensitic stainless steel (MSS) 410 deposits produced by laser powder-directed energy deposition (LP-DED) additive manufacturing. The LP-DED MSS 410 deposits underwent post-heat treatment, which included austenitizing at 980 °C for 3 h, followed by different tempering treatments at the temperatures of 250, 600, and 750 °C for 5 h, respectively. The analyses of phase transformations and microstructural evolutions of LP-DED MSS 410 were carried out using X-ray diffraction, SEM-EDS, and EBSD. Vickers hardness and tensile strength properties were also measured to analyze the effects of the different tempering heat treatments. It revealed that the as-built MSS 410 has very fine lath martensite, high hardness of about 480 HV_1.0_, and tensile strength of about 1280 MPa, but elongation was much lower than the post-heat-treated ones. Precipitations of chromium carbide (Cr_23_C_6_) were most commonly observed at the grain boundaries and the entire matrix at the tempering temperatures of 600 °C and 750 °C. In general, the tensile strength decreased from 1381 MPa to 688 MPa as tempering temperatures increased to 750 °C from 250 °C. Additionally, as the tempering temperature increased, the chromium carbide and tempered martensite structures became coarser.

## 1. Introduction

Additive manufacturing (AM) is a digital manufacturing process that produces near-net-shape or final shape parts composed of complicated geometries using a heat source for layer-by-layer build-up. For metal AM processes, a laser-based heat source has been used for powder bed fusion (PBF) and direct energy deposition (DED) [1,2,3,4].

Martensitic stainless steel (MSS) 410 has received considerable attention because it possesses key advantages, such as high strength and hardness, good corrosion resistance, forming ease, low material cost, air hardening, and good weldability. Due to its superior performance, it has been used in steam turbine components, pump shafts, valves, and bearings [5,6,7,8,9]. However, it is necessary to control the strength, ductility, and inherent embrittlement of MSS 410 by means of austenitization and tempering heat treatment processes [10,11,12]. Typically, the austenitization heat treatment process is followed by quenching, which increases material strength by the phase transformation of austenite to martensite [13,14,15]. The austenite phase grows by heating above the austenite transformation start temperature, and holding a sufficient heating time increases the carbon content and diffusion rate into the austenite grain boundaries to preserve a uniform phase composition and then transforms to the martensitic structure by rapid quenching. As a result, it increases hardness and strength. However, its toughness and ductility are very low; that is, it is highly brittle, so tempering heat treatment is necessary to increase toughness and ductility while sacrificing hardness and strength somewhat. Tempering heat treatment includes low-temperature tempering, which is heat treatment at 100~250 °C, and high-temperature tempering, which is heat treatment at 600~750 °C. Low-temperature tempering is performed in application environments where hardness is important, and it relieves internal stress and reduces brittleness [16,17,18]. On the other hand, high-temperature tempering decomposes martensite and forms coarse precipitation carbides, making it applicable in environments where toughness and ductility are important. As mentioned above, the application and heat treatment processes of conventionally cast MSS 410 materials are well-defined and reported to be utilized in industrial needs [19,20].

Recently, Roy et al. [21] reported the scatter in mechanical properties, especially toughness. It is correlated to microstructural heterogeneity in MSS 410 fabricated using arc-based additive manufacturing due to the formation of δ-ferrite in the deposition body. Meanwhile, Zhu et al. [22] developed MSS 410 large-size block parts without the defect of δ-ferrite precipitation using appropriate processing parameters via cold metal transfer wire-arc additive manufacturing. Nezhadfar et al. [23] reported the microstructure and crystallographic texture of 17–4 PH stainless steel with laser powder-directed energy deposition in both non-heated and heat-treated conditions. They found fine ferrite grains along with lath martensite; however, coarse ferrite was observed along the grain boundaries.

However, there has been little 3D printing LP-DED additive manufacturing research available on MSS 410, and limited investigation regarding additive manufacturing δ-ferrite-free-MSS 410. In this study, we researched MSS 410 fabricated by LP-DED additive manufacturing to analyze the effects of the microstructures, residual stresses, retained austenite, and mechanical properties of LP-DED MSS 410 deposits before and after post-heat treatment. In addition, the phase transformation and carbide precipitation were also investigated.

## 2. Materials and Methods

### 2.1. Materials and Additive Manufacturing

The integrated LP-DED equipment of a CNC machining center assembled with Optomec’s LENS (Albuquerque, NM, USA) deposition head with a beam size of 800 μm was used for additive manufacturing of the MSS 410. The feedstock material of the MSS 410 powder was manufactured using the electrode induction gas atomization (EIGA) process by KOSWIRE (Seoul, Republic of Korea). For sample fabrication, powder with particle sizes of 45–150 μm was employed, as shown in Figure 1.

The chemical compositions of the presently developed powder and its deposition are indicated in Table 1, confirming compliance with the AISI 410 standard specification.

For the preliminary experiments as seen in Figure 2, we conducted a total of 36 trial tests with 5 groups to find an optimal condition before producing mechanical characterization samples. Figure 2a shows the deposition cubes with different experiment parameters according to laser power and powder feed rate. Here, the Q and K values refer to the powder feeder’s motor rpm and laser output value, respectively, and are input variables for controlling the powder feeding rate and laser output. H is the final target height set in CAM, in mm.

Figure 2b shows a set of five experiments on laser power and powder feed rate to achieve good density and dimensional accuracy in this study, as follows:1st Group: explored 12 conditions combining 4 different laser powers (K350, K400, K450, K500, respectively) and 3 different powder feed rates (Q7, Q8, Q9).2nd Group: 12 conditions were investigated again by re-conditioning 3 different powder feed rates (Q4, Q5, Q6) targeting cube height of 5 mm.3rd Group: explored only powder feed rates (Q5, Q6) precisely.4th Group: deposited the cube height of 5 mm again.5th Group: set and deposited the cube target height of 15 mm and investigated the validity of the optimal conditions.

Finally, depending on the change in Q and K values, the optimal process parameters were selected with excellent density (over 99.9%) and close to the target height. The actual powder feed rate and laser output were measured, and the results are shown in Table 2. AISI 1045 carbon steel with the size of W × L × H = 200 × 200 × 30 mm^3^ was used as a substrate to prevent deformation during the laser beam deposition.

The laser deposition strategy was developed with bidirectional scanning, alternating the laser scanning path by 90° layer by layer, to reduce the anisotropy associated with both microstructures and mechanical properties of the deposition planes as shown in Figure 3.

### 2.2. Post-Heat Treatment

The prepared LP-DED MSS specimens with a size of W × H × L = 18 × 18 × 150 mm^3^ underwent post-heat treatments to characterize their microstructure, phase transformation, and mechanical properties compared with the as-built samples, as seen in Figure 4. The specimens were austenitized at 980 °C for 3 h at a heating rate of 2.3 °C/min, followed by forced fan cooling down to 100 °C at a cooling rate of 22 °C/min, then tempered at 250, 600, and 750 °C, respectively, for 3 h followed by furnace cooling.

### 2.3. Characterization of As-Built and Post-Heat Treatment Samples

For the characterization of the LP-DED MSS 410, the post-heat treatment samples were mainly cut into 3 pieces for the tensile tests, microstructures and X-ray diffractions, and hardness, as seen in Figure 5a. X-ray diffraction (XRD) analyses of as-built and post-heat treatment samples were carried out using an in situ X-ray diffractometer (EMPYREAN, Malvern Panalytical Co., Malvern, UK) with Cu Kα radiation. A continuous mode with a step size of 0.013°, scan rate of 0.05°/s, 2θ ranging from 30° to 90°, and power of 30 mA, 40 kV was applied to investigate phase transformation on the samples. The LP-DED samples were also cut into planes parallel (XY) and vertical (YZ) to the longitudinal deposition direction (X) for microstructural observation, as seen in Figure 5b.

The metallographic samples for SEM-EDS were prepared cautiously by polishing with a 1 μm diamond suspension and then etching with an aqueous solution (100 mL C_2_H_5_OH + 5 mL HCl + 3 g C_6_H_3_N_3_O_7_). Subsequently, electrolytic polishing was conducted in a solution (90 mL Methyl alcohol + 10 mL perchloric acid) at 28 volts for 90 s to achieve a refined surface state for electron backscatter diffraction (EBSD) analysis. The microstructures were characterized by SEM-EDS and EBSD. Vickers microhardness measurements were performed using a digital microhardness tester with a 1.0 kg load for 10 s dwell time on the polished planes parallel and perpendicular to the longitudinal deposition direction, in order to investigate the anisotropic texture structure disposition caused by laser scan direction-originated solidification of the melt pool. For the reliability of the tensile tests, five specimens were tested for each material condition. Tensile tests for the as-built and post-heat treatment samples were performed at a crosshead speed of 1 mm/min and at room temperature, in accordance with the specifications of round bar specimen No. 3 in the ASTM E8 standard using a universal test machine (Intstron 100 kN).

## 3. Results and Discussion

### 3.1. Phase Analysis

#### 3.1.1. Phase Equilibrium

Figure 6 shows the binary Fe-C phase diagram and phase fraction of LP-DED MSS 410 with 12 wt.% Cr and maximum carbon content of 0.2 wt.%, which were composed by Thermo-Calc^TM^ software calculating with the input chemical composition of the deposition, shown in Table 1.

When the LP-DED MSS 410 deposition rapidly transforms at temperatures between about 900 and 1100 °C from above the liquidus temperature at the carbon content of 0.13 wt.%, the LP-DED MSS 410 material is subject to transform in a trice to austenite through several phase transformations from the liquid state. Solidification continuously and rapidly proceeds. Martensite is predominant and consists of ferrite and carbide. Meanwhile, the chromium carbide (Cr_23_C_6_) starts to precipitate at the temperature of 950 °C and becomes stable below 800 °C.

#### 3.1.2. X-ray Diffraction Analysis

The XRD analyses of as-built and LP-DED MSS 410 samples with different tempering treatments after austenitizing at 980 °C for 3 h are shown in Figure 7. The crystalline LP-DED MSS 410 diffraction peaks were identified as the three major martensite peaks with a BCC structure (110), (200), and (210) by the good agreement of JCPDS cards of 96-901-3476 (Fe 2.00) and 96-901-4057 (Fe 4.00). These X-ray diffraction peaks are also in good agreement with Lu et al. [24]’s research. The MSS 410 feedstock powder has the lowest peak intensity due to the rapid quenching from the typical electrode induction gas atomization process, where argon gas blows the metal melt [25]. It is well known that the high-cooling rates of the DED AM process of 10^2^~10^3^ K/s are attributed to the peak intensities, depending on the process parameters, especially the scan speed [26,27].

It was revealed that there were some residual stresses in the samples of as-built and QT-250, which indicate the broad peaks. In particular, the retained austenite existed, as seen in thon as-built and QT-250. For the as-built sample, the residual stress and the retained austenite are largely attributed to the rapid quenching during additive manufacturing. After tempering at 600 and 750 °C, the residual stresses disappeared and became sharp peaks, releasing the residual stresses. Moreover, the retained austenite was not observed.

### 3.2. Microstructural Investigation

#### 3.2.1. SEM-EDS Analysis

Figure 8 shows the SEM images of LP-DED MSS 410 with different post-heat treatments. Lath martensite and incredibly small carbide precipitation, shown with the red arrows in Figure 8e,m, were seen in as-built microstructures. It implies that the phase transformation was developed through the rapid solidification sequence of the phase: L → L + γ → γ + M_23_C_6_ → α + γ + M_23_C_6_.

The microstructures of tempering at 250 °C illustrate the lath martensite and initial growth tempered martensite, showing a somewhat uniform distribution of tiny carbides within the developed lath martensite, as seen in Figure 8b,f,j,n. However, the lath martensite microstructures gradually decomposed after tempering heat treatment at 600 °C, and the tempered martensite grew coarser than that of tempering at 250 °C. In addition, most carbides, illustrated by red arrows, aggregated and precipitated within the lath boundaries and along the prior austenite boundaries over the tempered martensite, and the irregular and long-spun carbide size increased, as shown in Figure 8c,g,k,o. Compared to the microstructures of tempering at 600 °C, more carbides with somewhat spherical shapes, illustrated by red arrows, predominantly precipitated in the vicinities of the tempered martensite boundaries. Almost the same carbide precipitates within laths as well as along with lath boundaries after tempering at 923 K were reported by Chakraborty et al. [10]

Figure 9 shows the analyses of the energy-dispersive X-ray spectroscopy (EDS) on the precipitates with different tempering temperatures. The base material of LP-DED MSS 410 was confirmed as the main chemical compositions of Fe, C, Si, Mn, Cr, and Ni, as seen in Table 1. Due to rapid cooling during the LP-DED process, only tiny carbides were observed in the as-built specimen. However, in the QT specimens that underwent austenitization and tempering heat treatment, coarser carbides were observed. The higher the tempering temperature, the coarser the carbides that were observed. Although the Cr peaks are not large enough due to the small size of the carbide, it is inferred that the carbides formed after the tempering heat treatment were all M_23_C_6_ type. In particular, during low temperature tempering at 250 °C, both Fe_2_C and M_23_C_6_ carbides would possibly be formed as described by Chakraborty et al. [10]. As the tempering temperature increases, the diffusion of Cr increases; and Cr, which has a higher affinity for C compared to Fe, reacts to mainly form M_23_C_6_ carbide. A similar observation was researched with tempering at 732 °C for AISI martensitic stainless steel 410 reported by Godbole et al. [28].

#### 3.2.2. EBSD Analysis

EBSD phase maps, image quality (IQ) maps, inverse pole figure (IPF) maps, and kernel average misorientation (KAM) maps of perpendicular (YZ) planes of the as-built, QT-250, QT-600, and QT-750 samples are shown in Figure 10. In the as-built specimen, about 0.7% of retained austenite was observed, but after austenitization and tempering treatment, the amount of retained austenite in the QT-250, QT-600, and QT-750 samples decreased to 0.1%, as shown in Figure 10a. The decrease in the amount of retained austenite results from carbide formation, which occurs when the supersaturated carbon within the martensite diffuses out to form carbides during the tempering treatment. The increased size of lath martensite after tempering treatment can be confirmed qualitatively through the IQ map. The increased values can be quantitatively confirmed through the IPF map in Figure 10c. Compared to the as-built specimen, the size of prior austenite in QT-250, QT-600, and QT-750 has increased after austenitization heat treatment, and the size of lath martensite showing a misorientation within 15° has also increased in the QT specimens compared to the as-built. Additionally, it can be confirmed that the KAM value, i.e., residual strain, is relieved as the tempering temperature increases after austenitization heat treatment, as shown in Figure 10d. No evidence of δ-ferrite was observed in the KAM map.

### 3.3. Mechanical Property Characterization

#### 3.3.1. Hardness Test

Figure 11 shows the Vickers hardness of LP-DED MSS 410 samples with different post-heat treatments. It was revealed that the anisotropies of the Vickers hardness between parallel- and vertical-to-the deposition were observed in the samples of as-built and QT-250. It is apparent that these anisotropies are considerably attributed to the microstructural anisotropies, as shown in Figure 8. During depositing the MSS feedstock powder with the high scanning speeds of 1000 mm/min, the melt pool quickly solidifies onto the substrate layer-by-layer. As a result, the solidification rates are different from the top layer to the bottom layer. It was a very good trial to set the hardness test condition with the greater load of 1 kg, rather than 300 g as usual, to validate the anisotropies of microstructures and mechanical properties as well. After austenitizing at 980 °C and tempering the sample at 250 °C, these anisotropies gradually decreased, but the lower tempering temperature could not dissolve the deposition layers to uniform ones by recrystallizing the unstable and anisotropy microstructures. The isotropy, however, became prevalent after tempering at 600 and 750 °C, bringing the fully tempered microstructure with the homogeneous and coarse carbide precipitates in the vicinities of the prior austenite grain boundaries and lath boundaries. As a result, the reduction in the Vickers hardness was accompanied by the tempering temperatures which activate the nucleation and recrystallized grain growth incorporated with the Arrhenius equation as below.
(1)$$k=Ae^{\frac{-Ea}{RT}}$$
whereby *k* is the rate constant, *T* is the absolute temperature, *A* is the pre-exponential, *Ea* is the activation energy, and *R* is the universal gas constant.

#### 3.3.2. Tensile Test

Figure 12 shows the tensile properties of the LP-DED MSS 410 samples with different post-heat treatments. The tensile properties of the LP-DED MSS 410 samples with different heat treatment are summarized in Table 3. The yield strength, tensile strength, and elongation of as-built samples are 1109 MPa, 1281 MPa, and 19%, respectively. These as-built tensile properties have slightly higher values compared to the wire-arc-based AISI 410 additive manufacturing performed by Roy et al. [21]. It seems that the fine grains and residual stress caused by rapid solidification during additive manufacturing lead to an increase in tensile strength. However, the as-built and QT-250 samples show comparable values in tensile strength.

This can be attributed to recovery processes during the relatively low tempering treatment at 250 °C within the martensite formed through fan cooling subsequent to an austenitization treatment. This treatment leads to a reduction in dislocation density, resulting in a slight decrease in yield strength. It is noteworthy that the Vickers hardness of QT-250 perpendicular to the deposition direction was greater than that of as-built, which can have a higher tensile strength. Compared to QT-750, the tensile property of QT-600 was greater due to a finer grain size of 8.5 ± 6.7 μm and a higher residual strain value of KAM = 1.17°, as shown in Figure 10. Generally, the softening of the mechanical properties with tempering temperatures would be essentially attributed to the concentration increment of the chromium carbide precipitates along with grain boundaries.

## 4. Conclusions

Martensitic stainless steel 410 samples were produced by laser powder-directed energy deposition and then the samples underwent various post-heat treatments. The following conclusions can be drawn from this study:(a)The three major martensite peaks with a BCC structure (110), (200), and (210) were observed in all of the samples of as-built, QT-250, QT-600, and QT-750.(b)Some residual stresses and retained austenite were observed in the samples of as-built and QT-250 by X-ray diffraction analysis and EBSD.(c)The lath martensite microstructures decomposed after a tempering treatment of 600 °C and the tempered martensite grew coarser. In addition, most carbides aggregated and precipitated in the vicinities of the prior austenite grain boundaries and lath boundaries.(d)The contents of carbon and chromium increased as the tempering temperature increased, which is more favorable to form the carbide of Cr_23_C_6_, rather than that of Fe_2_C.(e)The decrease in the amount of retained austenite results from carbide formation, which occurs when the supersaturated carbon within the martensite diffuses out to form carbides during the tempering treatment.(f)The anisotropies of the Vickers hardness between parallel- and vertical-to-the deposition were observed in the samples of as-built and QT-250. These anisotropies are considerably attributed to microstructural anisotropies.(g)The fine grains and residual stresses caused by rapid solidification during additive manufacturing lead to an increase in tensile strength in the as-built sample compared to the different post-heat treatments.

## Figures and Tables

**Figure 1 micromachines-15-00837-f001:**
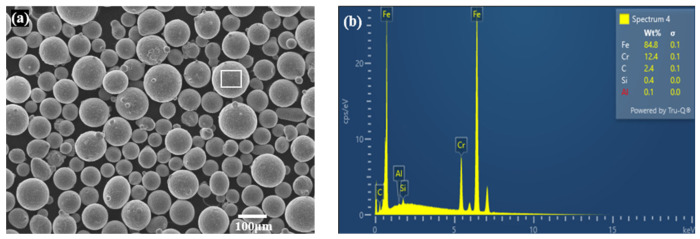
Micrographs of (**a**) feedstock MSS 410 powder and (**b**) EDS analysis.

**Figure 2 micromachines-15-00837-f002:**
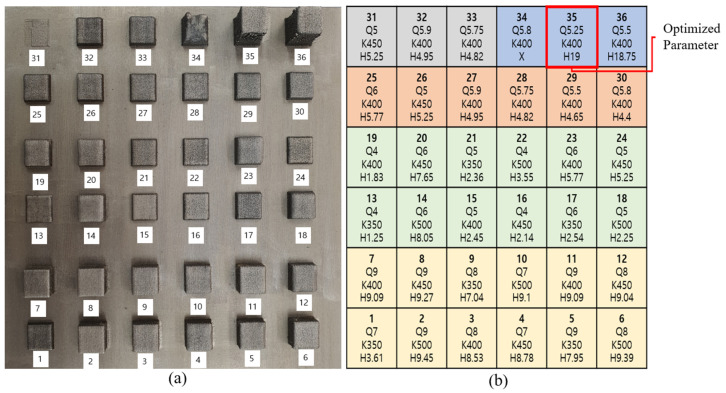
Deposition tests for optimal process development: (**a**) deposition cube pictures and (**b**) deposition parameters.

**Figure 3 micromachines-15-00837-f003:**
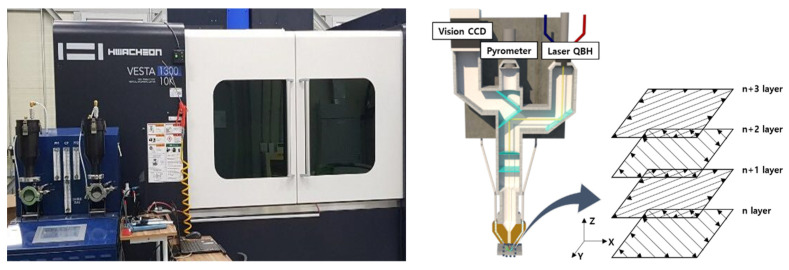
Schematic illustration of directed energy deposition scan strategy.

**Figure 4 micromachines-15-00837-f004:**
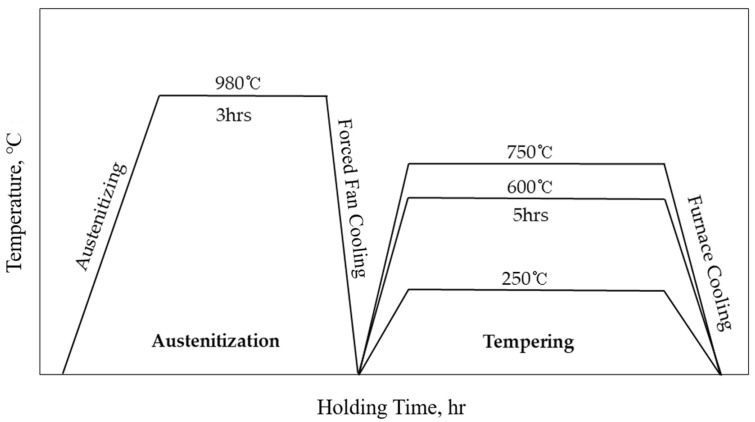
Schematic diagram of post-heat treatment for the LP-DED MSS 410 specimen.

**Figure 5 micromachines-15-00837-f005:**
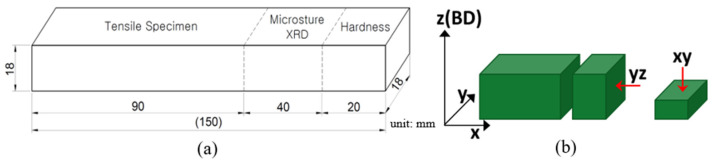
Schematic diagram of (**a**) overall cutting plan and (**b**) metallographic observation.

**Figure 6 micromachines-15-00837-f006:**
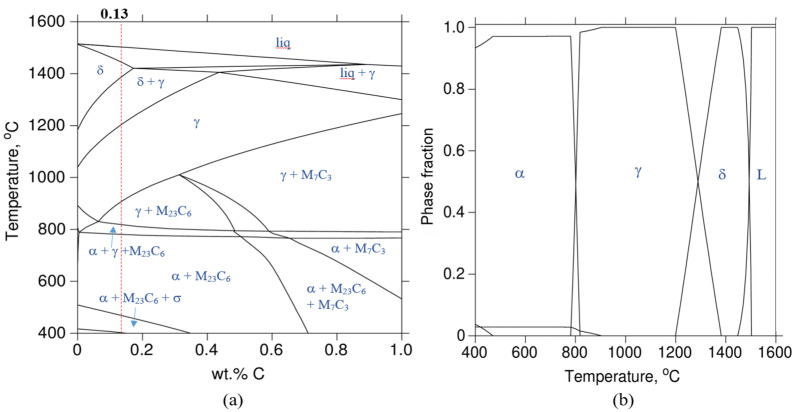
(**a**) Binary Fe-C phase diagram of MSS 410 constitution with 12 wt.% Cr and (**b**) its phase fraction: δ-Ferrite; α-Ferrite; γ-Austenite; σ-Sigma phase.

**Figure 7 micromachines-15-00837-f007:**
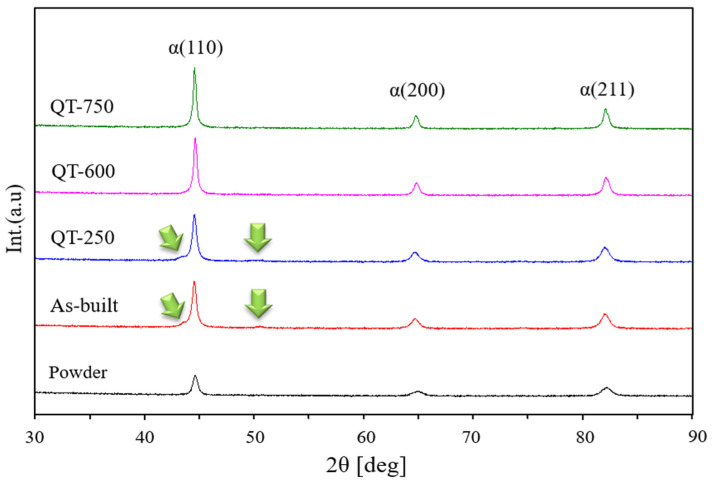
XRD patterns of LP-DED MSS 410 with different post-heat treatments: QT-250 is austenitized at 980 °C for 3 h and tempered at 250 °C for 5 h; QT-600 is austenitized at 980 °C for 3 h and tempered at 600 °C for 5 h; QT-750 is austenitized at 980 °C for 3 h and tempered at 750 °C for 5 h, as seen in Figure 4.

**Figure 8 micromachines-15-00837-f008:**
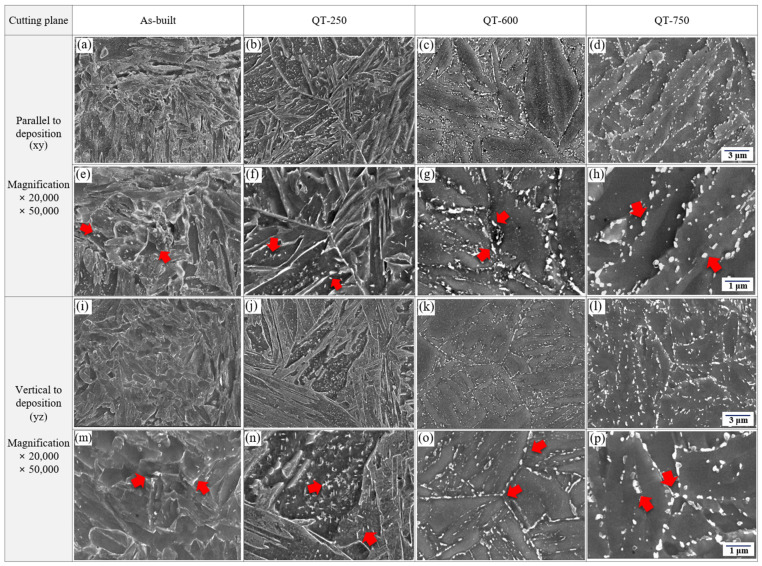
SEM images of LP-DED MSS 410 with different post-heat treatments: (**a**,**e**,**i**,**m**) as-built, (**b**,**f**,**j**,**n**) QT-250, (**c**,**g**,**k**,**o**) QT-600, (**d**,**h**,**l**,**p**) QT-750.

**Figure 9 micromachines-15-00837-f009:**
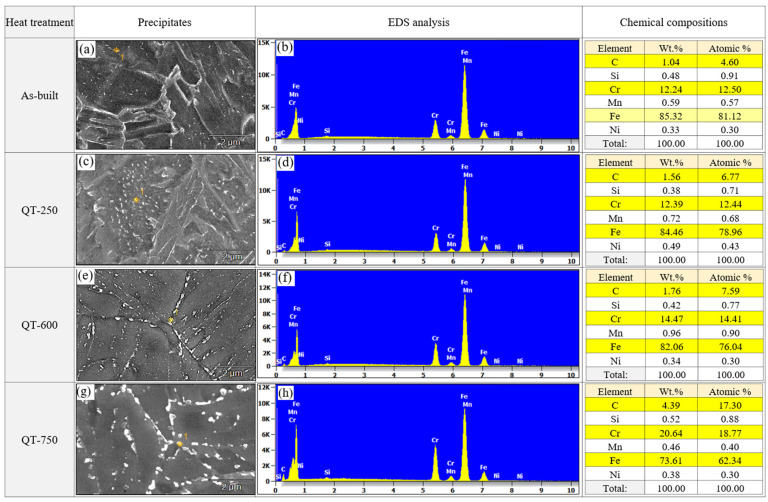
SEM-EDS analyses of LP-DED MSS 410 with different post-heat treatments: (**a**,**b**) as-built, (**c**,**d**) QT-250, (**e**,**f**) QT-600, (**g**,**h**) QT-750.

**Figure 10 micromachines-15-00837-f010:**
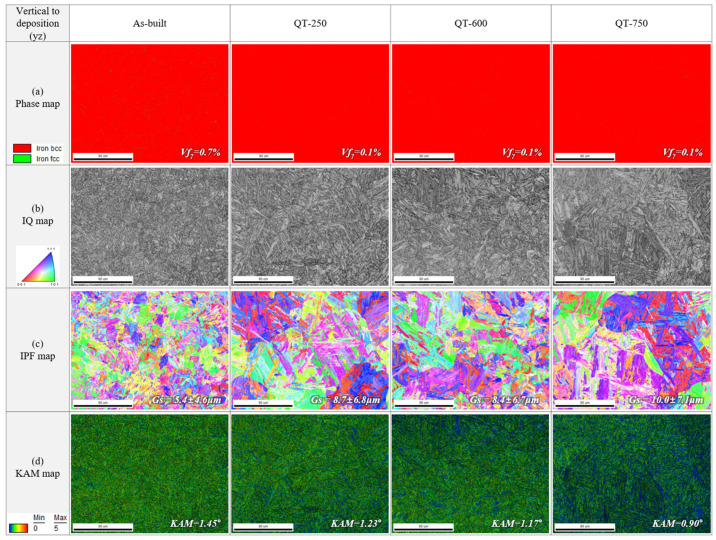
EBSD (**a**) phase maps, (**b**) image quality (IQ) maps, (**c**) inverse pole figure (IPF) maps, and (**d**) kernel average misorientation (KAM) maps of perpendicular (YZ) planes of the as-built, QT-250, QT-600, and QT-750 samples.

**Figure 11 micromachines-15-00837-f011:**
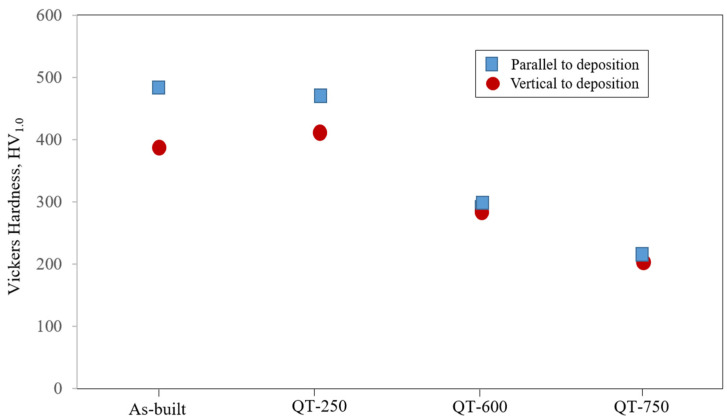
Vickers hardness of LP-DED MSS 410 with different post-heat treatments.

**Figure 12 micromachines-15-00837-f012:**
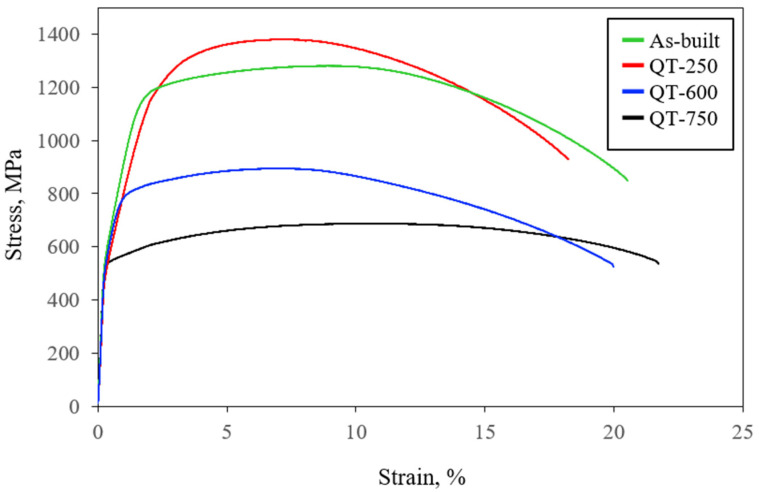
Tensile properties of LP-DED MSS 410 with different post-heat treatments.

**Table 1 micromachines-15-00837-t001:** Comparison of AISI 410 and the presently studied material (wt.%).

	C	Si	Mn	P	S	Ni	Cr	Fe
AISI 410	0.08–0.15	max. 1	max. 1	max. 0.04	max. 0.015	max. 0.6	11.5–13.5	Bal.
Powder	0.15	1	1	0.04	0.03	0.5	12	Bal.
Deposition	0.13	0.31	0.59	0.027	0.007	0.43	12.1	Bal.

**Table 2 micromachines-15-00837-t002:** Deposition parameters of the LP-DED MSS 410.

Process	Parameter	Value
LP-DED	Laser power (W)	600
	Scanning speed (mm/min)	1000
	Powder feed rate (g/min)	18.7
	Hatch distance (mm)	0.5
	Layer thickness (mm)	0.3

**Table 3 micromachines-15-00837-t003:** Comparison of tensile properties of the LP-DED MSS 410 samples.

	Yield Strength (MPa)	Ultimate Strength (MPa)	Elongation (%)	Reduction (%)
As-built	1109	1281	19	63
QT-250	1027	1381	17	60
QT-600	780	895	20	68
QT-750	542	688	22	52

## Data Availability

The data that support the findings of this study are available from the corresponding author upon reasonable request.

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
