# Peer review of "Microstructure and Mechanical Properties of Laser Direct Energy Deposited Martensitic Stainless Steel 410"

_micromachines, 2024, doi:10.3390/mi15070837_

Round 1

Reviewer 1 Report

Comments and Suggestions for Authors

Please consider the following modification suggestions:

1. In the introduction section, please increase the number of literature and research objectives.

2. Figure 5a should increase the unit in millimeters.

3. Suggest adding the unit degree to the horizontal axis of Figure 7.

4. Detailed image titles should be added to Figures 8 and 9.

Comments on the Quality of English Language

Minor editing of English language required

Author Response

Reviewer’s comments:

Reviewer #1:

Please consider the following modification suggestions:

  1. In the introduction section, please increase the number of literature and research objectives.

(Answer 1) The number of literature and research objectives was increased as shown in introduction.

  1. Figure 5a should increase the unit in millimeters.

(Answer 2)

The unit in millimeters was added in the Figure 5a.

  1. Suggest adding the unit degree to the horizontal axis of Figure 7.

(Answer 3) The unit degree was added in Figure 7.

  1. Detailed image titles should be added to Figures 8 and 9.

(Answer 4)

 Thank you for the comment. To improve readability, detailed figure captions are added as below.

  • Figure 8. SEM images of LP-DED MSS 410 with different post heat treatments: (a,e,i,m) as-built, (b,f,j,n) QT-250, (c,g,k,o) QT-600, (d,h,l,p) QT-750.
  • Figure 9. SEM-EDS analyses of LP-DED MSS 410 with different post heat treatments: (a,b) as-built, (c,d) QT-250, (e,f) QT-600, (g,h) QT-750.

Reviewer 2 Report

Comments and Suggestions for Authors

Review report

on the manuscript entitled “Microstructure and mechanical properties of Laser Direct Energy Deposited Martensitic Stainless steel 410” by authors Hyun-Ki Kang, Hyungsoo Lee, Chang-Seok Oh, Jongcheon Yoon (Manuscript ID: micromachines-3059490)

This work is focused on the examination of the microstructure and mechanical properties of laser-direct energy-deposited martensitic steel 410. It was found that the material in the as-built condition exhibited an almost completely martensitic structure with uniformly distributed M23C6 nano-scale dispersoids. The post-processing heat treatment (which involved austenizing annealing and subsequent tempering) resulted in microstructural coarsening and preferential precipitation of secondary particles along martensite grain boundaries. This resulted in the material softening.

In my opinion, this is an interesting and useful work that is worthy of sharing with the scientific community. Prior to its publication, however, I recommend the authors address several issues listed below.  

(1) In my opinion, the preferential concentration of secondary precipitates along grain boundaries during post-processing heat treatment may essentially contribute to the revealed material softening.

(2) The measured change in the chemical composition of the secondary particles during annealing treatment may be only an apparent effect. Due to the limited spatial resolution of the SEM-EDS techniques (~1 micron), the measured chemical composition of fine secondary particles is often “contaminated” by the chemical elements of the surrounding matrix material. The particle coarsening eliminates this “contamination effect”, thus resulting in an apparent change in the particle composition.

(3) Please discuss the EDS accuracy during the measurement of carbon content.

(4) It would be useful to detail the additive manufacturing conditions shown in Fig. 2b. Please also explain how the optimal regime was selected (porosity, hardness, sample appearance, etc.).

(5) Please indicate the number of tested tensile specimens per material condition.

Author Response

Reviewer #2:

This work is focused on the examination of the microstructure and mechanical properties of laser-direct energy-deposited martensitic steel 410. It was found that the material in the as-built condition exhibited an almost completely martensitic structure with uniformly distributed M23C6 nano-scale dispersoids. The post-processing heat treatment (which involved austenizing annealing and subsequent tempering) resulted in microstructural coarsening and preferential precipitation of secondary particles along martensite grain boundaries. This resulted in the material softening.

In my opinion, this is an interesting and useful work that is worthy of sharing with the scientific community. Prior to its publication, however, I recommend the authors address several issues listed below.

  1. In my opinion, the preferential concentration of secondary precipitates along grain boundaries during post-processing heat treatment may essentially contribute to the revealed material softening.

(Answer 1)

The opinion was reflected in the LINES 325-327.

  1. The measured change in the chemical composition of the secondary particles during annealing treatment may be only an apparent effect. Due to the limited spatial resolution of the SEM-EDS techniques (~1 micron), the measured chemical composition of fine secondary particles is often “contaminated” by the chemical elements of the surrounding matrix material. The particle coarsening eliminates this “contamination effect”, thus resulting in an apparent change in the particle composition.

(Answer 2) Thank you for the comment. The authors agree with the reviewer’s comment on the limitation of accuracy on SEM-EDS results. EDS analysis is readily to be operated rather than other composition analysis tools (WDS, ICP, etc.), but has the disadvantage of not being quantitative. Therefore, EDS results are mainly used for qualitative comparisons. Additionally, due to resolution limitations of EDS analysis, it is difficult to accurately obtain the composition of nm-scale carbide shown in Figure 9. Most of the x-rays are emitted from the matrix located below the carbide, so it is not easy to reveal the carbide composition by EDS. Therefore, it can be said that the higher carbon composition of carbide in QT-750 compared to as-built is mainly due to the carbide size. The manuscript has been revised as follows to reflect the reviewer's comment as below.

  • Revise manuscript LINES 235-248

Figure 9 shows the analyses of the energy-dispersive X-ray spectroscopy (EDS) on the precipitates with different tempering temperatures. The base material of LP-DED MSS 410 was confirmed as the main chemical compositions of Fe, C, Si, Mn, Cr, Ni as seen in Table 1. Due to rapid cooling during the LP-DED process, only tiny carbides were observed in the as-built specimen. However, in the QT-specimens that underwent austenitization and tempering heat treatment, coarser carbides were observed. The higher the tempering temperature, the coarser the carbides were observed. Although the Cr peaks are not large enough due to the small size of the carbide, it is inferred that the carbides formed after the tempering heat treatment were all M23C6 type. Particularly during low temperature of 250°C tempering, both Fe2C and M23C6 carbides would possibly be formed confirmed by Chakraborty et al. [10]. As the tempering temperature increases, the diffusion of Cr increases, and Cr, which has a higher affinity for C compared to Fe, reacts to mainly form M23C6 carbide. A similar observation was researched with tempering at 732℃ for AISI martensitic stainless steel 410 by Godbole et al. [28].

  1. Please discuss the EDS accuracy during the measurement of carbon content.

(Answer 3) Thank you for the comment. Analyzing carbon through EDS is extremely difficult due to the following reasons: low detection sensitivity of carbon element (low atomic number of carbon), carbon contamination during sample preparation, changes in carbon composition due to electron beam irradiation, etc. Therefore, the manuscript has been revised to mention qualitative differences and changes rather than mentioning quantitative numbers of carbon content. We thought that it was not suitable to include the discussion of the EDS accuracy of carbon content measurement, so we earnestly ask the reviewer to consider that this is not included in this manuscript.

  1. It would be useful to detail the additive manufacturing conditions shown in Fig. 2b. Please also explain how the optimal regime was selected (porosity, hardness, sample appearance, etc.).

(Answer 4) Refer to Lines 111-113

Depending on the change in Q and K values, the optimal process parameters were selected with excellent density (over 99.9%) and close to the target height.

  1. Please indicate the number of tested tensile specimens per material condition.

(Answer 5) For the reliability of the tensile tests, five specimens were tested for each material condition LINES 162-163.

Reviewer 3 Report

Comments and Suggestions for Authors

The manuscript investigated the properties of MSS 410 steel produced by AM for as-built samples and samples with post heat treatments. This manuscript can be considered for publication after some minor revisions.

In page 3, the parametric study of the AM process is not presented very well:

1)     The definition of Q, K, H are not clear. For example, what feed speed does Q7 corresponds to? Why is there also Q5.75, Q5.8, Q5.9 in Figure 2b, which are not mentioned in the text? Are these parameters decided by the authors or measured by the authors?

2)     For the 2nd group (I assume it’s the green color in Figure 2b), why the H ranged from 1.83 to 8.05 when the authors claim the height is 5mm in the text?

3)     It’s a bit confusing why the authors separate the samples into these five categories, and what is the logic behind for designing the parametric studies in each category?

4)     Most importantly, why the authors think No. 35 has the optimal property? What criterion do the authors use to judge its quality?

In page 9, please provide more details about Equation (1). For example, what is R, what is E, and what is a?

In page 10, can the authors comment on the effect on the brittleness/ductility from different tempering temperature of the samples?

Comments on the Quality of English Language

The English quality of the manuscript also needs to be improved. There’re a lot grammatic errors, unclear or colloquial expressions in some part of the manuscript. Please carefully revise.

Some of the places I spotted:

1)     Line 17. “was [much] lower than…”

2)     Line 27. “which [produces] a…”

3)     Line 35. “However, [it is] required to control [the] strength, ductility, [and the] inherent embrittlement [of the MS 410] by means of…”

4)     Line 42. “…to preserve ... and then [transform] to”

5)     Line 51, “As mentioned [above, the application and heat treatment processes of the conventional cast MSS 410 material] are …”

6)     Line 55. Please don’t use “And” at the beginning of a sentence. You either connect it to the previous sentence with a comma, or start a new sentence without "And". There are similar mistakes in other parts of the manuscript.

7)     Line 64. “there have been [few] 3D printing…”

8)     Line 73. For the unit micrometer, use “μm” instead of “um”. There are similar mistakes in other parts of the manuscript.

9)     Line 92. “we conduct [a total of] 36 trial tests… ”

10) Line 108. I think the authors mean: “steel with [the size of W x L x H = 200 x 200 x 30 mm3]”. For volume, use “mm3”. There are similar mistakes in other parts of the manuscript.

11) Line 130. Remove the red underline in the figure. There are similar mistakes in other figures.

12) Line 188. “…, [where] argon gas blow…”

13) Line 198. “retained austenite [existed]”

14) Line 234. “Compared to the as-built[, for the sample] austenitizing at …”

15) Line 235. “the chemical composition [slightly] increased to …”

16) Line 240. “as [described in] Chakraborty et al.”

17) Line 243. “[with] gradually decreasing iron”

In some part of the manuscript, especially section 3.2, the authors use very long sentences, which are hard to read. I suggest to break the long ones into short sentences less than three lines.

There’re also multiple places where a proper definite article should be used. Please carefully check.

Author Response

Reviewer #3:

The manuscript investigated the properties of MSS 410 steel produced by AM for as-built samples and samples with post heat treatments. This manuscript can be considered for publication after some minor revisions.

  1. In page 3, the parametric study of the AM process is not presented very well:
  • The definition of Q, K, H are not clear. For example, what feed speed does Q7 corresponds to? Why is there also Q5.75, Q5.8, Q5.9 in Figure 2b, which are not mentioned in the text? Are these parameters decided by the authors or measured by the authors?

(Answer 1-1) Thank you for your sharp advice. Q, K and H have been defined and the content has been modified in the text. To further explain, it is as follows.

  • Revise manuscript LINES 95-100, 109-11

The Q value and K value refer to the powder feeder's motor rpm and laser output value, respectively, and are input variables for controlling the powder feeding rate and laser output. H is the final target height set in CAM, in mm.

  • For the 2nd group (I assume it’s the green color in Figure 2b), why the H ranged from 1.83 to 8.05 when the authors claim the height is 5mm in the text?

(Answer 1-2) Thank you for the good advice. When the CAM S/W was set to a cube height of 5mm, the measured height (H) ranged from 1.83 to 8.05mm depending on the laser power and powder supply speed. To facilitate understanding of the content, the text has been modified as follows.

  • Revise manuscript LINES 103-104

12 conditions were investigated again by re conditioning 3 different powder feed rates (Q4, Q5, Q6) targeting cube height of 5 mm

  • It’s a bit confusing why the authors separate the samples into these five categories, and what is the logic behind for designing the parametric studies in each category?

(Answer 1-3)  The division into categories like this represents a set of experiments to evaluate whether the target height is approached and whether the density is excellent through a wide range of process conditions and subtle changes in process conditions. 

  • Most importantly, why the authors think No. 35 has the optimal property? What criterion do the authors use to judge its quality?

(Answer 1-4) The criteria for selecting the optimal process parameters were selected through proximity to the target height and measurement of the density of the deposited cube. This content has been corrected and added to the main text.

  • Revise manuscript LINES 111-113

Finally, depending on the change in Q and K values, the optimal process parameters were selected with excellent density (over 99.9%) and close to the target height. The actual powder feed rate and laser output were measured, and the results are shown in Table 2.

  1. In page 9, please provide more details about Equation (1). For example, what is R, what is E, and what is a?

(Answer 2) k is rate constant, T is absolute temperature, A is pre-exponential, Ea is activation energy, R is universal gas constant.

  • Revise manuscript LINES 299-300

Ea is activation energy, R is universal gas constant.

  1. In page 10, can the authors comment on the effect on the brittleness/ductility from different tempering temperature of the samples?

(Answer 3) Comparing as-built and QT-250, as-built shows a martensite matrix structure formed by rapid cooling in the ded lamination stage, and QT-250 is reverse transformed back to austenite during autenitization heat treatment, and then grain growth occurs, forming prior austenite. Not only the grain size but also the size of lath martensite showed an increased microstructure compared to as-built (8.7 μm vs. 5.4 μm in Figure 10c). Therefore, although both have the same microstructure made of martensite, there are differences in grain size and dislocation density, which causes the yield strength of QT-250 to decrease compared to as-built. However, since it can accommodate more dislocation multiplication, it shows higher work hardening compared to as-built, resulting in higher tensile strength. The factor that determines elongation is determined by how easily dislocations glide and cause slip. As-built specimens exhibit a higher density of high angle grain boundaries due to the smaller grain size (lath martensite), resulting in movement of dislocations. This can be interpreted as effectively delaying fracture compared to the QT-250 specimen, resulting in a higher elongation rate. In addition, looking at the change in hardness, it can be seen that the anisotropy of the QT-250 specimen decreased compared to the as-built specimen as the reverse transformation to austenite, that is, recrystallization, occurred through austenitization heat treatment. In QT-600 and QT-750, growth of prior austenite occurs primarily through austenitization heat treatment in the same way as QT-250. Compared to the QT-250 specimen, the QT-600 specimen that underwent tempering heat treatment at a higher temperature showed no additional grain growth, but the dislocation density (KAM value) decreased due to additional recovery (Figure 10d), and carbon diffused out from the lath martensite. As carbides precipitate along the lath boundary, the hardness and strength of martensite itself decrease, and the yield/tensile strength decreases. As the QT-750 specimen underwent a higher heat treatment of 750°C, additional grain growth occurred (Figure 10c) and it showed a very low dislocation density (KAM) value compared to the previous specimens (Figure 10d). In addition, compared to QT-600, carbon diffuses more actively, precipitating coarser carbides, and at the same time, the hardness of martensite itself is further reduced, changing into a soft martensite structure. Therefore, it shows the lowest yield/tensile strength and at the same time the highest elongation

  1. The English quality of the manuscript also needs to be improved. There’re a lot grammatic errors, unclear or colloquial expressions in some part of the manuscript. Please carefully revise. Some of the places I spotted:

1) Line 17. “was [much] lower than…” done

2) Line 27. “which [produces] a…” done

3) Line 35. “However, [it is] required to control [the] strength, ductility, [and the] inherent embrittlement [of the MS 410] by means of…” done

4)  Line 42. “…to preserve ... and then [transform] to” done

5) Line 51, “As mentioned [above, the application and heat treatment processes of the conventional cast MSS 410 material] are …” done

6) Line 55. Please don’t use “And” at the beginning of a sentence. You either connect it to the previous sentence with a comma, or start a new sentence without "And". There are similar mistakes in other parts of the manuscript. done

7) Line 64. “there have been [few] 3D printing…” done

8) Line 73. For the unit micrometer, use “μm” instead of “um”. There are similar mistakes in other parts of the manuscript. done

9) Line 92. “we conduct [a total of] 36 trial tests… ” done

10) Line 108. I think the authors mean: “steel with [the size of W x L x H = 200 x 200 x 30 mm3]”. For volume, use “mm3”. There are similar mistakes in other parts of the manuscript. done

11) Line 130. Remove the red underline in the figure. There are similar mistakes in other figures. done

12) Line 188. “…, [where] argon gas blow…” done

13) Line 198. “retained austenite [existed]” done

14) Line 234. “Compared to the as-built[, for the sample] austenitizing at …” revised

15) Line 235. “the chemical composition [slightly] increased to …” revised

16) Line 240. “as [described in] Chakraborty et al.” revised

17) Line 243. “[with] gradually decreasing iron” revised

In some part of the manuscript, especially section 3.2, the authors use very long sentences, which are hard to read. I suggest to break the long ones into short sentences less than three lines.

There’re also multiple places where a proper definite article should be used. Please carefully check. 

(Answer 4) Thank you for your kind advices. I checked.